# Neutrophil Functional Heterogeneity and Implications for Viral Infections and Treatments

**DOI:** 10.3390/cells11081322

**Published:** 2022-04-13

**Authors:** Lily Chan, Solmaz Morovati, Negar Karimi, Kasra Alizadeh, Sierra Vanderkamp, Julia E. Kakish, Byram W. Bridle, Khalil Karimi

**Affiliations:** 1Department of Pathobiology, Ontario Veterinary College, University of Guelph, Guelph, ON N1G 2W1, Canada; lchan12@uoguelph.ca (L.C.); vanderka@uoguelph.ca (S.V.); jkakish@uoguelph.ca (J.E.K.); 2Division of Biotechnology, Department of Pathobiology, School of Veterinary Medicine, Shiraz University, Shiraz 71557-13876, Iran; s.morovati@shirazu.ac.ir; 3Department of Clinical Science, School of Veterinary Medicine, Ferdowsi University of Mashhad, Mashhad 91779-48974, Iran; n.karimi@mail.um.ac.ir; 4Department of Pharmaceutical Sciences, University of Illinois at Chicago College of Pharmacy, Chicago, IL 60612, USA; kaliza2@uic.edu

**Keywords:** heterogeneity, plasticity, neutrophil, subsets, viral infection

## Abstract

Evidence suggests that neutrophils exert specialized effector functions during infection and inflammation, and that these cells can affect the duration, severity, and outcome of the infection. These functions are related to variations in phenotypes that have implications in immunoregulation during viral infections. Although the complexity of the heterogeneity of neutrophils is still in the process of being uncovered, evidence indicates that they display phenotypes and functions that can assist in viral clearance or augment and amplify the immunopathology of viruses. Therefore, deciphering and understanding neutrophil subsets and their polarization in viral infections is of importance. In this review, the different phenotypes of neutrophils and the roles they play in viral infections are discussed. We also examine the possible ways to target neutrophil subsets during viral infections as potential anti-viral treatments.

## 1. Introduction

Neutrophils are innate leukocytes with a complex immunobiology that serve many roles in immunity. Neutrophil heterogeneity is multifactorial, in which environmental cues can influence the phenotype and function of this cell type [1]. This includes inflammatory processes such as those that arise during viral infections. Neutrophils have been demonstrated to have both positive and negative effects for the host during immunological processes, which can be partially attributed to their heterogeneity and the stimuli they receive [2].

Due to constant exposure of the human body to external invaders as well as internal threats [3], leukocytes interact with their surrounding micro-environment and, as a result, modify their phenotype and function, which can include modulation of proliferation, differentiation, and cell migration capacity. We showed that exposure of oncolytic vesicular stomatitis virus to murine neutrophils affects their trafficking patterns, phenotype, and antigen-presentation potential in vivo [4]. The same may apply to human neutrophils during inflammatory responses [5,6,7] that alter their phenotype and functions, including metabolic and transcriptional reprogramming, which in turn, causes neutrophil heterogeneity under pathological circumstances [8]. Of interest, human neutrophil heterogeneity under physiological conditions has been also reported [9], raising the question of whether the interplay of different micro-environmental signals can cause neutrophil heterogeneity or whether these subsets have different developmental origins [10]. What programs and dictates heterogeneity in neutrophil immunobiology could be either the outcomes of neutrophil responses to micro-environmental cues or a developmental stage in neutrophil maturity. Single-cell analysis to dissect genetic alterations and molecular adaptability in response to changes in the local tissue microenvironment could define the drivers of phenotypic heterogeneity and functional versatility of neutrophils.

Through observing and understanding the ways in which neutrophil subpopulations arise and the mechanisms involved in controlling their functional and phenotypic plasticity during viral infections, it should be possible to determine methods to treat viral diseases that involve neutrophils. This would require understanding fundamental neutrophil biology, the various subsets, and how their functions contribute to both anti-viral protection and viral pathogenesis within specific immunological contexts. Although emphasizing the shortcomings of mouse-based studies of neutrophils when translating findings to humans, this review discusses the foundational experiments with murine neutrophils that facilitated breakthroughs in our understanding of human neutrophil immunobiology.

## 2. Neutrophil Immunobiology

Previously, neutrophils have been viewed as a relatively homogenous population of innate leukocytes that are terminally differentiated, short-lived cells with well-conserved functions. However, this idea has been challenged; rather than being a uniform population, neutrophils have been demonstrated to develop into multiple subsets with functional heterogeneity in response to physiological and pathological conditions such as inflammation, age, obesity, pregnancy, and even macro-environmental seasonality. Pillay and colleagues stated that despite the previous reports estimating a half-life of 8–10 h [11,12], the longevity of circulating neutrophils can increase to up to 5.4 days in humans and 3.75 days in mice [13]. Many host-dependent environmental cues and external stimuli can increase the longevity of neutrophils.

In a healthy state, neutrophil precursors undergo differentiation and development in the bone marrow and are released as mature segmented neutrophils. However, during inflammatory neutrophilia, the number of immature band neutrophils increases in the blood. Circulating neutrophils are drawn toward sites of inflammation or infection by locally released chemotactic agents such as constituents of a pathogen and chemoattractants secreted from damaged tissues or infiltrating leukocytes, including tumor necrosis factor (TNF)-*α*, interleukin (IL)-1, and IL-8 [14]. These cells show a distinct phenotype characterized by higher expression of adhesion and pathogen recognition molecules as well as proteases, namely, elastase [15]. Furthermore, the binding of integrin *α*9*β*1 expressed on neutrophils with vascular cell adhesion molecule-1 delays apoptosis of tissue-infiltrated neutrophils through activation of the nuclear factor-κB pathway [16]. The primary mode of recruitment of neutrophils into tissues is via attachment of these cells to proteins expressed on the surface of endothelial cells lining blood cells followed by extravasation. Integrins such as CD11a/CD18 and CD11b/CD18 expressed on neutrophils have sialylated Lewis X antigens, also referred to as cluster of differentiation (CD)-15 antigens, which interact with P-selectin and E-selectin adhesion molecules and intracellular adhesion molecule (ICAM)-1 expressed on endothelial cells [17,18,19,20]. Subsequently, neutrophils leave the circulation and enter the target tissues using CD11a/CD18, CD11b/CD18, and CD31 [21]. Neutrophil influx into the extravascular spaces leads to a phenotypic change from ICAM-1^low^,CXCR1^high^ to ICAM-1^low^,CXCR1^low^ [22].

Activation of neutrophils can serve as a link between innate and adaptive immunity and modulate immune responses. Interactions between leukocytes can occur directly or be mediated though cytokine secretion. Macrophages and natural killer (NK) cells can be activated and recruited to a site of infection upon the release of neutrophil-dependent pro-inflammatory factors such as IL-17, interferon (IFN)-*γ*, macrophage inflammatory protein (MIP)-1*α*, and MIP-1*β* [23,24,25,26,27,28]. In turn, activated macrophages and NK cells mutually extend the longevity of tissue-resident neutrophils through the secretion of granulocyte-macrophage colony-stimulating factor (GM-CSF), granulocyte colony-stimulating factor (G-CSF), TNF-*α*, and IL1β [29,30,31]. 

Contrasting with traditional views of unidirectional neutrophil infiltration from the bloodstream into extravascular tissues, new mouse studies show that neutrophils can undergo reverse transendothelial migration from tissues into circulation and even back to the bone marrow [22,32,33,34]. Interestingly, Tanshinone IIA, a compound that promotes neutrophil reverse migration, was identified in a zebrafish screen [35] and human neutrophils have also been shown to undergo reverse transendothelial migration in vitro [22] suggesting that this phenomenon may happen in mammals [36]. However, further investigations are required to fully explore reverse transendothelial migration in humans, particularly to determine if it occurs in vivo. The recirculating mouse neutrophils display a pro-inflammatory ICAM-1^high^CXCR1^low^ phenotype, and may contribute to systemic inflammation [33]. Furthermore, the lifespan of the neutrophils undergoing reverse transendothelial migration is prolonged through inhibition of apoptosis. Also, the cytotoxic potential of this subset of neutrophils is enhanced by the formation of neutrophil extracellular traps (NETosis) and production of reactive oxygen species (ROS) [22]. These findings should encourage future investigations to examine the occurrence of reverse transendothelial migration of neutrophils in humans and, if confirmed, discover medications to treat chronic inflammation by targeting this phenomenon.

Mouse neutrophils migrate to secondary lymphoid organs through lymphatic vessels. Therefore, they can interact with components of the adaptive immune system by shuttling invading microorganisms and/or presenting antigens. Neutrophils can provide antigen-presenting cells (APCs) such as dendritic cells (DCs) with intact pathogens or components thereof. They can also trigger maturation of DCs by producing factors that stimulate DCs, such as alarmins and TNF-*α* [37,38]. The reverse-migrated neutrophils display a specific phenotype of CD11b^high^, CD62L^low^, CXCR2^low^ in draining lymph nodes of mice. Indeed, they also display a high concentration of major histocompatibility complex (MHC) class II, CD80, and CD86, thereby allowing them to play a direct antigen presentation role within lymphoid organs [39]. During activation of DCs, human neutrophils bind to DC-specific intercellular adhesion molecule-3-grabbing non-integrin (DC-SIGN) through macrophage-1 antigen (Mac-1) and carcinoembryonic antigen-related cell adhesion molecule 1 (CEACAM-1) in vitro [40,41]. This physical interaction can also lead to the maturation of DCs by increasing the expression of several proteins—such as the epitope-expressing human leukocyte antigen (HLA)-DR and co-stimulatory molecules CD40, CD80, and CD86 [42]—and delays apoptosis of neutrophils. In mice, this immunological cross-talk has been demonstrated to enhance CD4^+^ T cell and B cell responses [43]. 

Interestingly, sex differences and seasonal changes can influence the recruitment and effector function of neutrophils. In mice, it seems that males are more susceptible to the detrimental effects of neutrophils that can occur during inflammatory responses, such as those induced by infection with group B streptococcus [44]. Furthermore, human neutrophils appear to provide more protective immunity in summer than winter. The phenomenon of how and why seasons influence immune function is an active area of research. There are several theories being explored, including the changes in the length of days and photoperiod, which are associated with melatonin [45]. Correspondingly, there has been observed regulations of neutrophils by melatonin in humans, mice [46], and zebrafish [47], which could account for the seasonal changes of neutrophil functional capacities.

As a part of the immune system’s first line of defense, neutrophils play a series of variable antimicrobial roles including phagocytosis, degranulation, NETosis, as well as respiratory bursts with the production of ROS. Phagocytosis is initiated by the opsonization of pathogens and the ligation of some receptors, such as antibody constant fragment (Fc)-*γ* receptors, C-type lectins, and complement receptors [48]. The phagosome is formed by neutrophils engulfing the pathogen, which takes place through the rearrangement of the cytoskeleton and changes in membrane proteins caused by cellular signals. This process can be enhanced by the complement cascade and IgG [49,50]. For instance, the complement cascade promotes the recruitment of phagocytes such as neutrophils by producing chemoattractants and also assists with the identification of pathogens, tagging them with opsonins that promote their phagocytosis [51]. Opsonins bind foreign materials and promote their phagocytosis; IgG can act as an opsonin and stimulate phagocytosis [52]. Once the phagosome is formed, primary and secondary granules inside neutrophils fuse with the phagosome and secrete their contents. The neutrophils of healthy individuals tend to be very heterogeneous in terms of phagocytic capacity, with some acting more efficiently than others [53].

The respiratory burst occurs along with phagocytosis. It results in the generation of ROS, namely superoxide anion, hydroxyl, and hydrogen peroxide as the result of nicotinamide adenine dinucleotide phosphate oxidase activation. In addition, myeloperoxidase (MPO) converts hydrogen peroxide to hypochlorous acid, which is another potent oxidant. These free radicals are also pumped into the phagosome to kill pathogens [54,55].

Degranulation or exocytosis is the release of the contents of neutrophils’ cytoplasmic granules into their surrounding environment. These can be divided into four classes:Primary or azurophil granules that are compacted with MPO, cathepsin G, elastase, proteinase 3, and defensins;Secondary or specific granules with lactoferrin as the most important component;Tertiary or gelatinase granules consisting of gelatinase proteins, such as matrix metalloproteinase (MMP)-9; andSecretory granules that mainly contain serum albumin and pre-formed cytokines [56].

Due to the toxic nature of some constituents of granules, degranulation is strictly regulated by two types of signals; the first is related to adhesion via *β*2-integrins, and the second depends on ligands and activated immunological receptors [57].

Another mechanism neutrophils employ for controlling pathogens is NETosis. NETosis is a form of programmed cell death where NETs—composed primarily of negatively charged neutrophil DNA and positively charged components, namely cytoskeletal proteins, elastase, histones, and some other toxic molecules—engulf and inactivate pathogens [48,58]. The oxidants generated by neutrophils degrade the nuclear envelope, resulting in DNA release from the nucleus into the cell. In addition, the enzymes peptidyl arginine deiminase IV and elastase contribute to DNA decondensation and nuclear envelope degradation [59,60]. Studies have shown that neutrophils do not necessarily die due to NETosis. This process can take place through vesicular transport and degranulation even independently of oxidants [61]. It should be noted that NETs could also cause damage to adjacent host cells, such as in some respiratory diseases, cancers, and autoimmune conditions, especially since they are covered with cytotoxic proteins and enzymes [62].

The signals affecting the choice between phagocytosis and NETosis have not yet been well characterized. However, some influential factors include environmental conditions, the magnitude of cell signaling in competing pathways, as well as the metabolic, adhesive, and activation state of the phagocyte. It has also been revealed that defective phagocytosis may bring about excessive NETosis, resulting in vascular inflammation, such as what occurs in systemic lupus erythematosus [63].

Under physiological conditions, neutrophils undergo apoptosis as they fulfill their function at inflamed sites. Indeed, these dead and dying neutrophils constitute a major component of pus that forms at some sites of inflammation. Apoptotic neutrophils are phagocytosed by tissue-resident macrophages, which limits inflammatory reactions. Apoptosis of neutrophils can occur through either intrinsic or extrinsic pathways accompanied by accumulation of pro-apoptotic B-cell lymphoma (Bcl)-2 family members and expression of high concentrations of proteins from the TNF receptor family [14,64,65]. However, under pathological conditions, clearance of neutrophils may be delayed, and these cells can undergo secondary necrosis. This causes the release of toxic contents from apoptotic neutrophils into the inflammatory micro-environment, thereby impairing the resolution of inflammation. In addition, structural damage to neutrophils can facilitate evasion of pathogens from neutrophil-mediated engulfment, which can lead to prolongation of the inflammatory process.

In mouse models, senescent neutrophils are characterized by an increase in concentrations of surface-expressed CXCR4 and a decrease in CXCR2-related responses. These changes enhance homing of neutrophils back to the bone marrow via the CXCR4 and stromal cell-derived factor-1 axis [66]. In vitro studies have shown that degranulation, respiratory bursts, chemotactic activity, and phagocytic activity of aged neutrophils are reduced compared to non-senescent neutrophils. The age and activation states of neutrophils also influence their localization and sequestration patterns. Although the spleen is considered to be a major site for elimination of circulating neutrophils, using mouse models, Suratt et al. [67] showed that removal of mature bone marrow and tissue-infiltrated neutrophils could occur via bone marrow stromal macrophages and liver-derived Kupffer cells, respectively. In vitro experiments have explored these pathways using human neutrophils to validate the relevance of the observations made in mouse models. Human neutrophils upregulate CXCR4 as they age and downregulate CXCR2, which makes the neutrophils more likely to engage with the CXCR4 ligand. It was observed that the younger neutrophils had increased migration towards IL-8—a ligand of CXCR2—compared to the aged neutrophils, while the older neutrophils were more inclined than the younger neutrophils to migrate towards stromal cell-derived factor-1—a ligand of CXCR4 [68]. These findings are in agreement with the previous studies showing that human aged neutrophils upregulate expression of CXCR4 and acquire the ability to migrate toward SDF-1*α* [69]. However, the findings require further investigation since Wolach et al. [70] demonstrated that CXCR4 expression in cultured human neutrophils occurs in the apoptotic cell population defined by Annexin-V staining, and that protein synthesis inhibition by cycloheximide blocked the upregulation of CXCR4 in vitro [71].

The effect of aging on the efficiency of innate immunity and subsequent risk of developing several diseases has been assessed in multiple experimental models. Contrary to initial studies, in which no changes or even decreases in neutrophil number were reported, recent investigations show the relation between high mortality rate in elderly individuals due to infectious diseases and increased neutrophil accumulation in infected tissues and circulation. While the neutrophils in elderly patients showed preserved adhesion function, their rolling along the endothelial wall was increased, thereby reducing leukocyte recruitment into tissues [72]. Also, neutrophils in the elderly tend to undergo premature apoptosis. Hence, their critical activities such as chemotaxis, phagocytosis, and microbicidal activity via free radical production are decreased, making elderly populations more susceptible to infections [73,74,75]. It seems that these functional age-dependent changes in neutrophils arise more from the alteration of the physicochemical status of the neutrophil plasma membrane rather than changes in receptor numbers [76,77]. This affects the functional properties of receptor signaling of lipid rafts and results in alteration of GM-CSF-mediated signal transduction, thereby decreasing the activation of Janus kinase (Jak)-signal transducer and activator of transcription (STAT), mitogen-activated protein (MAP) kinases, and the phosphoinositide 3-kinase (PI3K)-protein kinase B (PKB) pathways [77,78,79]. In addition to changes in membrane fluidity, defects in membrane calcium metabolism [80] and regulation of the actin cytoskeleton [81] impede neutrophil effector functions with aging. According to Butcher and colleagues [73], attenuation of Fc-mediated phagocytosis is related to expression of a remarkably lower concentration of CD16 on neutrophils. Also, higher expression of Bcl-2 associated X (Bax)/myeloid cell leukemia (Mcl)-1 also reduces the rescue of neutrophils from apoptosis in elderly subjects, which may contribute to alterations in the JAK2/STAT signaling pathway [82]. Furthermore, neutrophil extracellular activities such as NET formation [83] and their interaction with other components of the immune system, including DCs, macrophages, and lymphocytes are decreased with age, leading to impaired immune responses to pathogens [84]. Class switching of B cells [85], reduction of immunoglobulin (Ig) and complement protein concentrations [86], and shifting T cell phenotypes from a T_helper_ (Th)-1 towards Th2 bias [87] are some factors minimizing the cooperation between innate and adaptive immune systems in elderly populations.

## 3. Neutrophil Heterogeneity and Plasticity

Neutrophils exhibit phenotypic and functional plasticity in response to different physiological and inflammatory conditions that occur during viral infections. Heterogeneous populations of neutrophils are classified based on discrete features, including cell-surface markers, cell receptors, maturity, density, and functions [1]. It should be noted that defining and differentiating subsets of this highly heterogenous leukocyte is difficult for a multitude of reasons, including the plasticity between subsets, changes that occur as a cell matures, and that there are some subsets that only arise under specific circumstances. This makes it challenging to investigate subpopulations and define subsets. Transcriptomic, functional, density, and surface marker analysis are some methods researchers use to distinguish different neutrophil subpopulations [88]. As a result, the literature has many proposed definitions for subpopulations of neutrophils, and there is overlap between defined subsets and multiple names associated with some populations. However, understanding neutrophil heterogeneity and plasticity could provide insights to assist with the identification of neutrophil-related biomarkers of pathogenesis, determinants of disease, and development of treatment plans that account for the roles of neutrophils.

Current evidence indicates that neutrophils can be separated into subsets based on size and density. Blood density gradient centrifugation can separate mononuclear cells from granulocytes (Figure 1). Conventional neutrophils will be found in the granulocyte fraction and are sometimes referred to as normal-density neutrophils (NDNs), and low-density neutrophils (LDNs) are found in the mononuclear fraction. LDNs have been described to exert both pro-inflammatory and immunosuppressive effects, suggesting that LDNs can be further separated into subsets [89]. LDNs are typically populations associated with disease, and NDNs are considered to be the typical neutrophils that would be found in healthy individuals [89]. In patients with systemic lupus erythematosus (SLE), LDNs have been observed to express higher concentrations of CD11b and CD66b than NDNs, which are markers associated with cell activation [90,91]. LDNs and NDNs have also been isolated from the blood of healthy individuals and examined for phenotypic and functional characteristics. NDNs were observed to have lower expression of CD15 and CD16b relative to LDNs [92], which suggests LDNs and NDNs differ in functional capacities and can be distinguished from each other via surface markers.

Profiling granulocytic cells in cases of COVID-19 [93], neutrophils were defined as CD15^+^CD66b^+^CD193^−^CD16^bright/dim^ and their proportions increased as disease severity progressed. This increase was detected in the proportion of both mature (CD16^+^) and immature (CD16^−/dim^) neutrophils. In patients with COVID-19, high concentrations of CD66b, CD177, CD11b, CXCR4, CD147, and CD63, and low concentrations of CXCR2 were expressed on neutrophils. Patients with severe COVID-19 had higher proportions of immature neutrophils that had decreased expression of CD66b and CD11b compared to neutrophils associated with mild disease and healthy controls. The patients with moderate COVID-19 had higher proportions of mature neutrophils, with higher expression of CD11b, CD66b, and CD177 than those with severe COVID-19 and the controls. In patients with moderate COVID-19, they found increases in neutrophil populations expressing phenotypic characteristics that would support type 1 responses, such as CXCL-9, CXCL10, and IL-15. They also exhibited an increase in neutrophil populations that expressed interferon response-related molecules, IL-2, and B cell activating factor, which would suggest the neutrophils supported anti-viral and adaptive immune responses. Once recovered from COVID-19, the proportions of immature neutrophils that were observed to be elevated during COVID-19 decreased and were restored to levels comparable to the healthy controls [93]. This study illustrated the heterogeneity of human neutrophils in viral infections such as SARS-CoV-2 and the roles they could play in the pathogenesis or resolution of the diseases.

One study investigated neutrophils in humans that received G-CSF and explored CD10 expression and the influence on T cell responses. CD10 is a marker that assists in defining maturity, with the CD10^−^ and CD10^+^ phenotypes defining immature versus mature neutrophils, respectively. This is important for differentiating mature CD66b^+^ LDNs from immature CD66b^+^ LDNs, since they appear to differentially influence T cell responses [94]. It was observed that CD10^+^ neutrophils (both LDNs and NDNs) hinder T cell proliferation and IFN-*γ* production. Their obstruction of T cell responses appears to be dependent on a mechanism involving CD18 and the release of arginase I. However, CD10^−^ LDNs were shown to have the opposite effect on T cell responses and promoted proliferation, survival, and production of IFN-*γ*, but also via a mechanism involving CD18 [94].

LDNs can be further categorized based on maturity, with immature cells exhibiting a CD11b^−/low^CD16^−/low^ phenotype and mature LDNs being CD11b^+^CD16^+^, as observed in various human studies reviewed by Scapini et al. [90]. Activation usually results in downregulation of CD16. Therefore, it is difficult to differentiate immature neutrophils from those that are activated based on these markers alone [90]. Morrissey et al. [88] examined neutrophils in human patients with COVID-19 and separated LDNs into three groups based on expression of CD16. The concentration of expression of CD16 appears to correlate with morphological maturity, with low, intermediate, and high expression corresponding to an appearance similar to precursors, immature cells, and mature cells, respectively. They found that CD16^int^ LDNs had increased NETosis, phagocytosis, and degranulation and genes associated with a pro-inflammatory profile. They also found an association between CD16^int^ LDNs and plasma concentrations of the cytokines IL-10, IL-1RA, monocyte chemoattractant protein (MCP)-1, and MIP-1*α* [88].

Although LDNs were thought to appear only under inflammatory conditions, Blanco-Camarillo et al. [92] investigated LDNs in healthy individuals. They observed a population of LDNs found among the mononuclear cells isolated via density gradient centrifugation that had the phenotype CD10^+^, CD11b^+^, CD62L^+^, CD66b^+^, CXCR4^+^, CD15^high^, CD16b^high^, CD14^low^. This population was similar to NDLs in their ability to mediate NETosis and they demonstrated a higher capacity for phagocytosis and ROS production than NDLs [92]. This suggested that their existence did not require inflammatory stimuli. Instead, it appears that a subset of LDNs can exist under a physiological steady state and that these may have some enhanced functional capacities. Further investigation is required to clarify whether LDNs are present in healthy individuals, or, if as shown in more studies, they are a pro-inflammatory neutrophil subset that is associated with the severity of many immune-mediated inflammatory diseases.

Myeloid-derived suppressor cells (MDSCs) are a heterogeneous population of immature myeloid cell progenitors, which are activated under a large array of physiopathological conditions ranging from malignancies and chronic infections to pregnancy. They are divided into two main groups: mononuclear myeloid-derived suppressor cells (M-MDSC) and granulocyte myeloid-derived suppressor cells (G-MDSC), distinguished by distinct cell surface markers. Generally, in mice, G-MDSCs present with a CD11b^+^,Ly6G^+^,Ly6C^low^ phenotype, while the monocytic subset displays CD11b^+^,Ly6G^−^,Ly6C^high^ [95]. In humans, G-MDSCs are lower in density and CD11b^+^,CD15^+^,CD14^−^,CD33^+/low^,CD66b^+^, and M-MDSCs are CD14^+^,HLA^−^,DR^−/low^ [96].

There is another subset of immunosuppressive myeloid cells known as low-density polymorphonuclear myeloid-derived suppressor cells (PMN-MDSCs), which are also described in different pathological and homeostatic conditions. It should be noted that throughout the literature, there are differences in opinion on the definitions of PMN-MDSCs and G-MDSCs. There are some researchers that consider PMN-MDSCs and G-MDSCs as the same cell subset [96], while others suggest they are distinct cell subsets, or that PMN-MDSCs are a subgroup of G-MDSCs. Since G-MDSCs arise from myeloid precursors, there is heterogeneity amongst G-MDSCs attributed to the level of maturation of the precursor when it became a MDSC [97]. Therefore, it appears that PMN-MDSCs could be classified as a subset of G-MDSCs that arose from a more mature neutrophil precursor that had a segmented nucleus. The term ‘G-MDSCs’ appears to be more abundantly used throughout the literature. Regardless, they are both considered to be neutrophil-derived MDSCs due to their mutual similarities to neutrophils. In this review, PMN-MDSCs will be considered a subset of G-MDSCs.

These immunosuppressive cells are crucial negative regulators of T cell immunity, leading to persistent viral infection and other clinical complications, including tumor metastases. MDSCs, as non-differentiated cells, do not express mature myeloid markers such as HLA-DR2 and other lymphoid surface markers such as CD3, CD19, and CD56 [98]. The phenotypes of human MDSCs can vary based on physiological and pathological states and their anatomic site. However, they usually are defined as CD11b^+^ or CD33^+^ cells in humans [95,99] and GR1^+^ and CD11b^+^ cells in mice [100,101]. In steady-state conditions, immature myeloid cells differentiate into mature leukocytes. However, during pathological conditions, immature cells traffic to secondary lymphoid organs and incite their immunosuppressive activities. Neutrophilic MDSCs have been identified in patients with cancers [97,102] and tumor-bearing animals [103]; they are also associated with fungal [104,105] and bacterial [106,107] infections. The functions of these cells may correspond to worse outcomes or protect the host from invading pathogens. In addition, through the secretion of a variety of inflammatory and anti-inflammatory cytokines by MDSCs, they can influence the differentiation of different leukocyte subsets, including naïve CD4^+^ T cells, as well as the functionality of various leukocytes [95,108,109].

Cellular immunological dysfunction has been reported in people with chronic hepatitis C viral (HCV) infections [110]. This may in part be due to the remarkable accumulation of a subset of G-MDSCs that have an HLA-DR^−/low^, CD33^+^, CD14^−^, CD11b^+^ phenotype in the serum of infected individuals. The accumulation of G-MDSCs in the liver is a reliable criterion for the prognosis of patients with chronic HCV infections, as it can be used for survival prediction with respect to the development of hepatic cancers. Wang et al. [111] demonstrated the direct relationship between the concentration of circulating HCV core protein and the frequency of G-MDSCs. G-MDSC polarization occurs through activation of the extracellular signal-regulated protein kinase (ERK)-1/2 mitogen-activated protein kinase (MAPK) pathway, with further augmentation by IL-10-dependent signal transducer and activator of transcription (STAT)-3 phosphorylation. Eventually, the proliferation of autologous T cells and production of IFN-*γ* are repressed, leading to persistence of HCV.

During infections associated with severe pyrexia and thrombocytopenia syndrome virus, which as those caused by HCV, G-MDSCs downregulate the expression of the T cell CD3-ζ chain that is critical for the assembly of T cell receptors and their downstream signal transduction. Consequently, human studies have demonstrated that this impedes viral clearance that would normally be mediated through arginine metabolism by nitric oxide synthase and arginase [112,113]. Furthermore, mouse models have shown that arthritogenic alphaviruses such as chikungunya virus and ross river virus induce the accumulation of a unique myeloid cell population of macrophages and neutrophils in musculoskeletal tissues. These cells expressing arginase 1 delay disease resolution in an arginase 1-dependent pathway [114].

T cell anergy in people infected with human immunodeficiency virus (HIV) is strongly related to disease progression. Some scientists have found that M-MDSC subsets are expanded during chronic infection with HIV [115,116]. Zhang et al. [117] suggested that immunological environments and virus replication induce exhaustion of T cells by stimulating the production of HLA-DR^−/low^,CD14^−^,CD33^+^,CD11b^+^ granulocytic MDSCs in the initial phase of infection with HIV. These cells implement their inhibitory effects on CD8^+^ T cells through the programmed death protein (PD)-1/PD-1 ligand axis which leads to disease progression. The frequency of G-MDSCs is negatively correlated with expression of CD3ζ on T cells [118]. CD3ζ down-modulation is observed during chronic inflammation, which can occur in the context of cancers, autoimmune disorders, or infections. In patients infected with HIV, this immunosuppressive phenomenon is mediated by the silencing of the transcription factor E74-like factor (ELF)-1 [118].

According to Chen et al. [119], murine hepatic MDSCs with a CD11b^+^ Gr1^+^ phenotype impair hepatitis B virus (HBV)-specific cellular immunity through the suppression of HBV surface antigen-specific lymphocyte proliferation and expression of IFN-*γ*. The number of these cells in the livers of HBV-transgenic mice is twice that of wildtype mice. The immunosuppressive functions of CD11b^+^ Gr1^dim^ MDSCs exceeds that of their CD11b^+^ Gr1^high^ counterpart.

It is also documented that impaired T cell immunity during infection with chronic lymphocytic choriomeningitis virus is associated with the expansion of Gr-1^high^ neutrophilic cells in lymphoid tissues and blood [120]. Recruitment of myeloid progenitor cells is due to cytotoxic T lymphocyte-mediated lysis of infected hematopoietic cells during infection with persistently infecting strains of lymphocytic choriomeningitis virus; this is mediated by stimulatory molecules such as CCL2 [120].

The proportion of PMN-MDSCs is increased significantly in patients with infectious diseases associated with severe pyrexia with thrombocytopenia syndrome, as well as in patients with cancers. These cells play a fundamental role in feto-maternal immunological tolerance and angiogenesis through the suppression of proliferation of CD4^+^ and CD8^+^ T cells [121]. Lang et al. [97] investigated MDSC populations in patients with head and neck cancers and observed greater T cell suppression from PMN-MDSCs as compared to the other MDSC subsets, and the number of PMN-MDSCs correlated with poorer disease prognosis.

In humans, PMN-MDSCs have a phenotype similar to neutrophils based on being CD15^+^, CD66b^+^, CD16^+/−^, CD14^−^, CD11b^+/−^, CD33^+^, and HLA-DR^−^. However, no differentiating markers have been characterized that can group them into two distinct classes yet, as has been done in mice. Nevertheless, they can be distinguished from other neutrophils according to density gradients and gene expression profiles. For example, genes related to cell cycle, autophagy, and guanine nucleotide-binding (G)-protein-mediated signaling are upregulated in PMN-MDSCs, while mature neutrophils have higher expression of NF-κB and lymphotoxin-*β*-receptor signaling-related genes [122]. In this context in mouse models, the STAT3 transcription factor is upregulated in MDSCs and is essential for the accumulation of MDSCs in the bone marrow and spleen [123,124]. However, in humans and mice, it has been demonstrated that hypoxic tumor micro-environments decrease the expression of STAT3 in tumor-infiltrating MDSCs by triggering CD45 phosphatase [125]. PMN-MDSCs are responsible for more endoplasmic reticulum stress responses compared to other subsets of neutrophils. Furthermore, induction of endoplasmic reticulum stress can change the neutrophils to PMN-MDSCs that have a suppressive phenotype containing related endoplasmic reticulum stress sensors [122]. In patients with cancers, it has been demonstrated that PMN-MDSCs can be distinguished from other neutrophil populations by their expression of lectin-type oxidized low-density lipoprotein receptor-1 (LOX-1). LOX-1 association with PMN-MDSCs was shown to only occur in human neutrophils and was not seen in murine neutrophils. Although a growing body of evidence describes the PMN-MDSCs as a population of immature cells, some recent studies have pointed out that individuals with systemic lupus erythematosus and various cancers have mature neutrophils that express high concentrations of cell surface-expressed CD10 [94].

In addition to cytotoxic T lymphocyte-mediated immunity, PMN-MDSCs also subvert CD4^+^ T cell responses in mouse models. The MDSC expansion following infection with Japanese encephalitis virus dampens the functionality of T follicular helper cells, thereby facilitating progression of disease. The suppression of T follicular helper populations subsequently leads to decreased numbers of splenic B cells, CD19^+^ CD138^+^ plasma cells, total IgM, and Japanese encephalitis virus-specific neutralizing Abs, thus impairing antibody-mediated immunity [126]. Additionally, G-MDSCs play a significant role in blocking NK cell functions during adenoviral infections in mice. They accumulate in the spleen where they mediate ROS-dependent suppression of NK cells [127].

A robust cytotoxic T lymphocyte response is ideal in human neonates with primary respiratory syncytial virus (RSV) infections [128]. Virus-dependent damage during severe primary RSV disease in neonates is most likely initiated by a systemic neutrophil influx into the blood and airways. Based on expression of CD16/CD62, four distinct functional subpopulations of immature, progenitor, mature, and suppressor neutrophils have been defined in infants with severe viral infections, such as those caused by RSV [128,129,130]. Immature and progenitor neutrophils have been characterized as having CD16^low^CD62L^high^ and CD16^low^CD62L^low^ phenotypes, respectively. Presumably, low expression of the CD62L extravasation marker in bone marrow-derived progenitor neutrophils may restrict them to certain anatomic areas. Although lymphocyte paralysis is observed in infants with a severe viral infection, the suppressive activity of the infant-derived neutrophils (immature and progenitor cells) has not been demonstrated yet. Nevertheless, suppressive neutrophils (CD16^high^CD62L^low^) are detected in pediatric patients with RSV and bacterial co-infections [129].

The complexity in neutrophil heterogenicity and plasticity is translated to their functional participation during viral infections. Neutrophils can contribute to protective immune responses against viruses, as well as participating in responses that ultimately result in damage to the host and exacerbate the viral infection (Figure 2). The dual nature of neutrophils in viral infections will be discussed in the next section. 

## 4. Function of Neutrophil Phenotypes

### 4.1. Protective Effects of Neutrophil Subsets during Anti-Viral Defense 

As discussed, neutrophils have complex heterogeneity, with varying functions in immunity. This section will examine neutrophils’ protective functions during viral infections. One of the more thoroughly researched functions of neutrophils is NETosis [58]. Chikungunya virus initiates non-lytic NETosis through a mechanism that involves Toll-like receptor (TLR)-7 activation and generation of ROS [131]. Murine and human NETs have been observed to capture and neutralize chikungunya virus and help control the infection [131]. During chikungunya virus infection in zebrafish, neutrophils have been found to produce type 1 IFNs, which are critical in the control of the infection [132]. Additionally, human neutrophils NETs have been documented to significantly decrease the ability of dengue virus to infect cells, further demonstrating their anti-viral capabilities [133]. NETosis can also be initiated indirectly in human neutrophils. For instance, during immune responses against viruses, IgA can directly neutralize viruses but can also stimulate certain leukocytes, including neutrophils, via Fc alpha receptor (Fc-αR)-I-dependent pathways [58]. IgAs can form immune complexes with viruses and engage Fc-αRI, which promotes a ROS-dependent suicidal NETosis in neutrophils. The NETs are then capable of trapping and inactivating viruses [58]. IgA can also significantly lower the threshold dose of a virus required to induce NETosis, therefore helping to limit disease progression [58]. These findings suggest NETs play a protective role in a variety of viral infections when properly regulated. However, their activation and mechanism of action may differ depending on the viral infection.

Neutrophils have also been shown to phagocytose virus particles. Following influenza vaccination of mice, the chemokine CXCL1 and the alarmin molecule IL-1α initiated rapid recruitment of neutrophils [134]. In the popliteal lymph node, neutrophils initially exhibited patrolling behavior followed by swarming and were confirmed to phagocytose and transport influenza virus particles [134]. Murine neutrophils’ ability to transport antigens indicates they may also present viral antigens to promote adaptive immunity. Murine neutrophils also partake in the activation of other innate leukocytes. Following respiratory viral infection, neutrophils were demonstrated to be critical in activating the NOD-, LRR-, and pyrin domain-containing protein (NLRP)-3 inflammasome in alveolar macrophages, committing them to a pro-inflammatory phenotype to aid in combating infection [135]. Neutrophils can also recruit lymphocytes. For instance, following infection of mice with influenza virus, neutrophils have been shown to leave a trail of chemokines, especially CXCL12, that guide CD8^+^ T cell migration to the infected tissues [136]. These examples of neutrophil crosstalk with other leukocytes demonstrate how neutrophils can indirectly influence the outcome of viral infections.

Although neutrophils are conventionally associated with innate immunity, they also play roles in the induction of adaptive immunity. One study investigated neutrophils’ capacity to act as antigen transporting cells and demonstrated the potential to elicit antigen-specific T cell responses during viral infection. The study utilized an attenuated strain of vaccinia virus expressing HIV-1 clade C antigens to infect mice [137]. Following infection, the NFκB pathway was activated, which led to the expression of several cytokines that increased the migration of neutrophil populations [137]. Di Pilato et al. [137] distinguished these populations by size and complexity in mice. They were named Nα and Nβ, with Nβ being larger, more lobulated, and more segmented than Nα. Both of these neutrophil populations have been described to have an in vitro capacity to generate antigen-specific CD8^+^ T cell responses after recruitment by cytokines associated with the pro-inflammatory environment following infection [138]. However, Nβ were observed to be more mobile and showed a greater capacity to act as an APC to activate virus-specific CD8 T cells [138]. They also expressed higher concentrations of the epitope-expressing molecule MHC class II and co-stimulatory molecules CD11, CD80, and CD86 than Nα, further supporting their enhanced ability to act as APCs compared to Nα [137,138]. These neutrophil populations are speculated to be able to transport antigens that have already been processed and released by apoptotic cells [137]. They can transport antigens to lymphoid organs—including the spleen and lymph nodes—to prime CD8^+^ T cells directly and promote antigen-specific immune responses [137,138].

Vono et al. [139] characterized the ability of human neutrophils to act as APCs. Neutrophils were pulsed with the cognate antigens pp65 from cytomegalovirus or influenza hemagglutinin and were able to present the antigens to autologous antigen-specific CD4^+^ T cells via MHC class II [139]. They exhibited this APC potential and could induce proliferation of helper T cells at a low but consistent rate both in vitro and ex vivo [139]. 

Further research into neutrophils’ functions in the promotion of adaptive immunity is of interest to fully understand their roles in anti-viral defense. For example, a study using samples from humans, mice, and rhesus macaques investigated a subset of neutrophils found in the marginal zone of the spleen, called B cell-helper neutrophils, which were found to increase IgM secretion by marginal zone B cells [140]. They were also shown to activate marginal zone B cells as effectively as splenic CD4^+^ T cells and more effectively than splenic macrophages and DCs [140]. Thus, investigating their capacity to assist in B cell responses during viral infections could be valuable for designing anti-viral therapeutic strategies.

MDSCs are subsets of neutrophils and monocytes that are pathologically activated and function to dampen immune responses [141]. Although inhibiting immune responses can have harmful effects and could be seen as counterintuitive for leukocytes to take part in, there are circumstances where it can be beneficial. MDSCs have potent immunosuppressive functions and can assist in controlling excessive immune responses that cause off-target damage and harm to the host [141]. An example of the protective function of MDSCs has been demonstrated in vivo in mouse models following infections with influenza viruses. Some influenza infections can cause exaggerated immune responses, such as those caused by T cells, and lead to disease progression [142]. One of the immunosuppressive neutrophil subsets, G-MDSCs, has been shown to suppress T cell-mediated pathology in this scenario [142]. Preliminary studies have also shown that MDSCs play a similar role in humans after infection with severe acute respiratory syndrome-coronavirus-2 (SARS-CoV-2), where they have been shown to dampen excessive immune responses by suppressing T cell functions [143]. Overall, neutrophils perform many different anti-viral functions. However, their activation, functional activity, and subsequent mechanisms must be tightly regulated.

### 4.2. Detrimental Potential of Different Neutrophil Subsets and Their Promotion of Collateral Damage during Viral Infections

Although neutrophils function to protect the host, they have been shown to contribute to the pathology of severe diseases. Neutrophilia, an increase in the number of neutrophils circulating in the blood, often correlates with the progression of viral infections and severity of the associated disease. Excessive neutrophil infiltration in the lungs is seen in cases of severe rhinovirus and influenza infections in mouse models [144] and humans [145,146]. They are implicated in the development of acute lung injury and acute respiratory stress disorder (ARDS) often resulting from these infections [144]. In mice, early neutrophil migration into the brain and cerebrospinal fluid characterizes viral encephalitis, a disease mostly associated with herpes simplex virus (HSV)-1 [147]. More recently, studies indicate that human neutrophil activation and degranulation are highly activated processes following infection with SARS-CoV-2 [148]. These results are enhanced by the knowledge that neutrophilia has been described as an indicator of severe respiratory symptoms and a poor outcome in the novel coronavirus disease that was first identified in 2019 (COVID-19) [149,150,151,152]. It has been observed that neutrophil count in the circulation was the greatest independent prognostic indicator for COVID-19 mortality [153], highlighting the importance of neutrophils in viral infections and the prognostic potential for neutrophils in COVID-19, which also has implications for treatment courses.

The roles of neutrophils after infection with RSV have been debated among researchers. It is known that the host immune response to infection is a major determinant of the severity of disease. Studies conducted in infants with severe RSV disease have implicated neutrophils in disease pathogenesis, often showing an increase in neutrophils and neutrophil elastase in the lungs [154,155]. McNamara and colleagues even demonstrated that neutrophils make up over 90% of the cellular compartment of the bronchoalveolar lavage [154]. However, studies also suggest that neutrophils do not contribute to disease pathogenesis in the context of RSV bronchiolitis. Kirsebom et al. [156] depleted the lungs of RSV-infected mice of neutrophils and saw no effect on disease severity, suggesting that in some inflammatory contexts, the potentially pathogenic effects of neutrophils can be tolerated or even controlled [156]. These results are not surprising as Emboriadou et al. [155] saw no significant correlation between neutrophil elastase and disease severity. In this way, RSV is a good example of a very important caveat in establishing the effects of neutrophil activity in viral infection. When there is an inability to establish causation, it is entirely possible that exacerbated neutrophil responses are a consequence of immune dysregulation that accompanies severe disease, rather than being the drivers of disease.

Neutrophils contribute to collateral damage during viral infections in several ways. The release of proteolytic enzymes such as elastase and MMP9 from neutrophils via degranulation causes host cell damage, inflammation, and degradation [157,158]. Viral infections are associated with a decrease in antioxidant defenses, leaving the immune system vulnerable to damage by ROS [159]. Through degranulation, neutrophils also produce ROS, exacerbating the immunopathological response and resulting in more severe disease [157,160]. In addition to the cellular damage caused by products released through degranulation, neutrophils contribute to airway obstruction by inducing mucus production [157]. All these processes are particularly destructive in delicate tissues such as the lungs, explaining why neutrophilia is implicated in the development of many severe respiratory diseases.

Using a mouse model of viral encephalitis, Micheal et al. [161] demonstrated the role of early neutrophil migration in the induction of increased blood brain barrier permeability, which is characteristic of viral encephalitis. By abrogating CXCL1 signaling, they observed a reduction in neutrophil recruitment, blood brain barrier permeability, and morbidity [161]. This implicated neutrophils as the key drivers of increased blood brain barrier permeability, morbidity, and mortality in viral encephalitis.

However, the most detrimental collateral damage of neutrophils may be caused by the process of NETosis. For example, although NETs have been seen to trap RSV particles, their formation is excessive and can lead to severe lower respiratory tract disease [162]. As with neutrophils, accumulation of NETs was involved in the obstruction of airways and direct injury to tissue in human and bovine studies [162,163]. NETs contribute to increasing the permeability of epithelial cells, impairing gas exchange, and promoting severe disease in mice and humans [164,165]. DNA released from these NETs entangle with the endothelium of small blood vessels, resulting in capillary damage [163]. Toussaint et al. [166] also demonstrated that in humans and mice, the double-stranded DNA released during NETosis contributes to the disease pathology of asthma that sometimes gets exacerbated following infections with rhinoviruses. By preventing the release of double-stranded DNA, the recruitment of pro-Th2 DCs to the lungs was impeded, thereby decreasing type 2 immune-mediated exacerbation of asthma [166].

Work by Radermecker et al. [167] demonstrated that by infiltrating distinct lung compartments, NETs contributed to different aspects of COVID-19 physiopathology. The increase in NETs in the airway compartment led to an increase in pro-coagulant factors, leading to fibrin deposition and subsequent impaired pulmonary ventilation [167]. NET-releasing neutrophils were also found in proximity to the interstitial compartment [167]. Neutrophils stimulated macrophages to release IL-1*β*, in turn increasing NET formation and IL-6 secretion, contributing to the detrimental cytokine storm observed in severe cases of COVID-19 [167]. NETs were also found in the vascular compartment of the lungs, localized in arterioles containing microthrombi, providing a scaffold for adhesion of platelets and molecules as well as erythrocytes [167]. These scaffolds demonstrated a way in which NETs participate in thrombus formation and exacerbate COVID-19 [167,168]. The neutrophils found in the vascular compartment were Cit^−^, H3^+^, and MPO^+^, representing a population of activated neutrophils in a relatively early stage of NET formation [167]. While the study was conducted on a small number of patients, this work highlights the importance of defining the way in which NETs can contribute to the physiopathology of a novel respiratory disease.

There is also ample evidence in humans that some viruses, including HIV-1 and RSV, directly contribute to the observed increase in NETosis following infection [162,169]. While the process of this stimulation has yet to be fully elucidated, using human and murine samples, Sung and colleagues found that dengue virus activates platelets via C-type lectin-like receptor (CLEC)-2, causing them to release extracellular vesicles that further enhance NET formation through activation of CLEC5A and TLR2 on macrophages and neutrophils [170]. Narasaraju et al. [144] also demonstrated that when incubated with influenza virus-infected alveolar epithelial cells, neutrophils were stimulated to undergo NETosis in mouse models. This was likely a result of increased secretion of CXCL-5, a chemokine known to stimulate neutrophils and induce chemotaxis [144].

In certain circumstances, increases in conventionally defined bulk neutrophils has been demonstrated to be an effective prognostic marker. Correspondingly, the expansion of MDSC populations can also be used to predict disease progression. The increase in MDSC subset numbers in response to viral infection appears to favor immunosuppression and viral persistence. An increase in both G-MDSC and M-MDSC subset populations are associated with disease progression following infection with HIV, with high concentrations predicting increased viral load and a decrease in the number of CD4^+^ T cells in patients with HIV [117,171,172]. Increased frequencies of G-MDSCs and M-MDSCs were found in patients infected with HBV, and increases in the total number of MDSCs and M-MDSCs were identified in patients infected with HCV [173,174,175,176,177]. Immunosuppression mediated by these cells could contribute to chronic infection and the development of the cancers often associated with these viruses, which are known as hepatocellular carcinomas [178,179]. This process is evidenced by the observed increase in MDSCs in patients with HBV- and HCV-related hepatocellular carcinomas, as compared to those with chronic infections [179,180]. Most recently, Reizine and colleagues demonstrated a critical role of MDSCs in the development of COVID-19-related ARDS [181]. MDSCs were increased in patients with ARDS compared to patients with moderate pneumonia, suggesting that MDSCs were responsible for the T cell dysfunction observed in COVID-19-related ARDS, which is the most severe form of COVID-19 [181].

Heterogeneity of MDSCs can also be observed in terms of the functional mechanisms through which these cells induce immunosuppression, most notably through the exhaustion of T lymphocytes. They can express several enzymes and molecules with suppressive function such as indoleamine-pyrrole 2,3-dioxygenase (IDO), arginase-1, and reactive nitrogen and oxygen species, and they can upregulate immunological checkpoints such as PD-L1 and galectin-9 [182]. In mouse models, CD39, an essential molecule involved in adenosine metabolism, has also been identified as a possible mechanism of MDSC-mediated suppression of CD8^+^ T cells [183]. There was also an observed decrease in MDSC suppression of CD8^+^ T cells in the absence of CD39 [183].

## 5. Implications of Neutrophil Heterogeneity for the Treatment of Viral Infections

An enhanced understanding of neutrophil heterogeneity and the different pathways employed by these cells to respond to diverse stimuli could introduce new therapeutic avenues for virus-mediated diseases. This could be particularly beneficial in the selective targeting of neutrophils given the dual role of these cells in viral clearance and the perpetuation of inflammation.

Acute inflammation can induce the production of G-CSF, which can affect two different subsets of neutrophils. First, it can recruit CD10^+^ mature neutrophils with immunosuppressive roles into human blood. In addition, it mobilizes CD10^−^ immature neutrophils that stimulate T cell survival [94]. It is also known that the immunomodulatory effects of neutrophils produced in response to the production of G-CSF and other myelopoietic factors are different from those recruited from other sources [10], giving rise to the functional heterogeneity of neutrophils during inflammation. These differences could be exploited at early and late stages of inflammation to selectively target the subpopulations of neutrophils. Moreover, the inhibition of G-CSF, which in turn influences its effects on diverse populations, could be taken into consideration. An investigation on periodontal inflammation using human and murine samples revealed that G-CSF-neutralizing antibodies significantly diminished neutrophil infiltration, the expression of some chemokines and interleukins, and associated tissue destruction [184]. Another approach to alleviate the impacts of G-CSF is to target its receptor. Campbell et al. [185] developed a monoclonal antibody against the murine receptor for G-CSF, which could antagonize the binding of G-CSF. They observed that neutrophil accumulation, adhesion receptors of these cells, and pro-inflammatory cytokines—such as IL-1*β* and IL-6, which are known to play roles in tissue damage—declined in inflamed joints. Moreover, the gene expression profile of neutrophils that remained following treatment differed from the inflammatory phenotype. It should be noted that they found no adverse effect on the clearance of influenza virus as the result of G-CSF receptor blockade [185].

LDNs are another potential target given that viral infections such as dengue virus [186], HIV [187], and SARS-CoV-2 [188] alter different phenotypes of LDNs in patients. For example, LDNs in patients infected with HIV showed increased expression of CD66b, CD63, and CD11b on their surface [187]. In patients with severe fever and thrombocytopenia syndrome, an emerging disease associated with viral infections, these cells secreted more pro-inflammatory cytokines than other populations [189]. Moreover, LDNs with a weakened oxidative burst response were reported in severe cases of COVID-19 [188], and their numbers remained elevated even 3 months after infection [190]. Another group of researchers also detected CD16^+^ LDNs in the blood and bronchoalveolar lavage fluid of patients with COVID-19, which were suggested to be associated with coagulopathy, systemic inflammation, and ARDS in these patients [88]. Interestingly, different populations of neutrophils have been shown to vary in terms of their ability to produce NETs. In particular, LDNs have been reported to be elevated in autoimmune diseases, such as systemic lupus erythematosus, with an enhanced capacity to form NETs [91,94,191]. Higher NETosis by LDNs, compared to their high-density counterparts, has also been found in cases with antiphospholipid syndrome; the authors concluded that neutrophils may augment the risk of thrombosis, thereby making them a target for preventing thrombosis in these patients [192]. If an association between LDNs and NETosis could be established in virus-related inflammation, the characteristic LDN phenotypes in such conditions can be used to design new anti-inflammatory measures specifically targeting these cells.

As mentioned, an increase in LDNs has been observed in patients with SLE. An in vitro study investigating NETosis in pediatric patients with SLE showed that mature neutrophils isolated from children with SLE were more prone to randomly die than neutrophils from healthy individuals. Neutrophils from patients with SLE underwent NETosis and death shortly after in vitro exposure to ribonucleoprotein-specific antibodies, which are induced in a proportion of patients. It was also observed that these SLE-associated neutrophils that undergo NETosis upon stimulation with anti-ribonucleoprotein antibodies induce activation of plasmacytoid DCs. This NETosis-induced plasmacytoid DC activation was also demonstrated in neutrophils of healthy individuals. However, they required stimulation with IFN-α before exposure to anti-ribonucleoprotein [193]. Therefore, this data suggests that neutrophils can promote type I IFN responses, which could be an indirect anti-viral mechanism of NETosis and has the potential to be explored therapeutically.

A study investigating immune suppression by neutrophils in the blood of patients infected with HIV-1 observed that neutrophils had increased expression of PD-L1 following infection. The PD-L1/PD-1 pathway is a mechanism for immunosuppression, especially contributing to T cell suppression. Increases in IFN responses, antigen-specific T cells, and CD8^+^ T cells producing IFN-*γ* were observed when CD15^+^ LDNs were depleted [194]. This suggests the potential to develop therapies by targeting immunosuppressive CD15^+^ LDNs. Furthermore, another study observed a correlation with aged neutrophils and immune suppression in the blood of patients infected with HIV-1 [195]. The study defined aged neutrophils as CXCR4^+^ and CD62L^low^ and non-aged neutrophils as CXCR4^−^ CD62L^+^. They also scored the age of the neutrophils using age-associated markers and a scoring system described in detail by Xie et al. [196]. Patients with HIV had increased aging scores for neutrophils, and this increase was associated with a decrease in CD4^+^ T cell responses. There was an association between aged neutrophils and expression of PD-L1. Patients with HIV that had not received anti-retroviral therapy had increased expression of PD-L1 and arginase-1 on neutrophils compared to healthy controls and patients that received anti-retroviral therapy. This indicates that as neutrophils age, they upregulate PD-L1 and arginase-1. Blood-derived mononuclear cells co-cultured with aged neutrophils had decreased CD8^+^ T cell responses compared to blood-derived mononuclear cells co-cultured with non-aged neutrophils, which suggests that aged neutrophils suppress T cell responses. Blood-derived mononuclear cells co-cultured with neutrophils from untreated patients with HIV-1 decreased the production of IFN-γ and TNF-α by CD8^+^ T cells, which was partially restored when arginase-1 or PD-L1 was blocked [195]. This supports the idea that patients with HIV-1 have neutrophils that suppress their T cell responses and suggests that a possible mechanism to reverse this suppression could be by targeting arginase-1 and PD-L1.

The population of LDNs with intermediate CD16 expression was increased in number relative to the other two LDN subpopulations in patients with severe COVID-19. It was proposed that infection with SARS-CoV-2 promoted the CD16^int^ LDN phenotype. There was a relationship between concentrations of TNF-*α* and IL-6 and intermediate CD16-expressing LDNs. These two powerful pro-inflammatory cytokines are proposed to be the main contributors to the cytokine storm associated with severe COVID-19 [88]. Therefore, a possible way to target and reduce these potent cytokines is by depleting this population of neutrophils that expresses intermediate CD16. There was a correlation between numbers of CD16^int^ LDNs and clinical outcome; therefore, the concentration of this subpopulation of neutrophils could also potentially be used as a biomarker and monitored throughout the course of infection to assist with determining prognosis and treatment plans. CD16^int^ neutrophils expressed high concentrations of CXCR3, which binds to CXCL10 and is involved in the localization of neutrophils. The paper suggested that the increased CD16^int^ neutrophils in the lungs could be attributed, at least in part, to the CXCL10/CXCR3 axis. Therefore, there is also the potential to target this axis to hinder recruitment of neutrophils and prevent SARS-CoV-2-induced immune-mediated damage in the lungs [88].

It has been reported that patients with COVID-19 have higher numbers of neutrophils that exhibit increased expression of IFN-related genes and enhanced induction of type 1 IFNs compared to neutrophils in control patients that do not have COVID-19. Treatment with the steroid dexamethasone reduced the number of neutrophils that expressed IFN-stimulated genes and promoted immunosuppressive immature neutrophils [197]. This could be explored as a potential avenue to attenuate damage associated with neutrophil-mediated pro-inflammatory responses involving IFN signaling during viral infections such as SARS-CoV-2.

Chemokine receptors on neutrophils are generally what control their recruitment, activation, and extravasation. Neutrophils can vary in their expression of these surface receptors, which largely dictate where they go and what they will do. Understanding which receptors do what and how to control their expression could lead to mechanisms of how to manipulate neutrophil localization and functional heterogeneity [198]. Rudd et al. [198] observed that neutrophils found in the lungs after infection with influenza virus had different chemokine receptors than neutrophils in circulation in mouse models. These receptors included CCR1, CCR2, CCR3, CCR5, CXCR1, CXCR3, and CXCR4. Of the neutrophils in circulation and those recruited to the lungs of healthy and infected individuals, CXCR2 was the most highly expressed among them. There was a larger frequency of neutrophils expressing all the receptors, except CXCR2, in infected lungs compared to blood from infected and uninfected patients. The neutrophils in the blood of infected individuals had a slightly lower frequency of expression of CCR1, CCR2, CCR3, and CCR5 [198]. Phagocytic activity of the neutrophils recruited to the lungs could be abrogated by blocking CXCR2 and CCR5. However, it was observed that an increase in phagocytic activity occurred after blocking CCR1. Therefore, the phagocytic activity of neutrophils recruited to the lungs during viral infections could be manipulated by blocking CXCR2, CCR5, and/or CCR1. Furthermore, neutrophils isolated from the lungs had increased migratory capacity and NET release following stimulation with IL-8, CCL3, and CCL4 [198]. Therefore, NET release could be promoted by using IL-8, CCL3, or CCL4, or obstructed by blocking IL-8, CCL3, CCL4, or the corresponding receptors.

Middleton et al. [199] examined the relationship between NETs and COVID-19, investigating the potential to inhibit NETosis using a NET-inhibitory peptide called neonatal NET-inhibitory factor. Neonatal NET-inhibitory factor was able to obstruct COVID-19-induced NETosis; therefore, targeting NETosis might be a promising therapeutic avenue for neutrophil-induced pathogenesis in some viral diseases, including COVID-19.

It has been demonstrated that inhibition of the protein gasdermin D hinders NETosis in neutrophils isolated from mice or humans [200]. Disulfiram is a drug used in the treatment of alcoholism due to its ability to inhibit aldehyde dehydrogenase. It has also been shown to inhibit gasdermin D in mice and humans, which is needed for NETosis [201]. One study showed that inhibition of gasdermin D using disulfiram hindered neutrophil NETosis and improved sepsis outcomes. This was performed using mouse models, and the clinical relevance was validated using neutrophils from human blood samples that demonstrated hindrance of NETosis by disulfiram via a mechanism that involves gasdermin D [201]. This concept was investigated further in a hamster model of SARS-CoV-2 infection and mouse models of transfusion-related acute lung injury by Adover et al. [202]. They demonstrated that disulfiram hindered Netosis, decreased neutrophil infiltration to the lungs, and helped prevent tissue damage during SARS-CoV-2 infection; however, it did not negatively influence viral clearance in a hamster model of COVID-19. In a mouse model of transfusion-related acute lung injury, treatment with disulfiram decreased NETosis and limited lung injury [202]. Therefore, there is potential to use gasdermin D inhibitors such as disulfiram as a therapeutic approach to prevent NETosis-induced damage by neutrophils during viral infections.

It has been suggested that infection with some respiratory viruses predisposes individuals to post-viral atopic diseases [203,204,205,206,207,208]. Accordingly, neutrophils are responsible for developing virus-mediated airway hyperreactivity. Given this, Cheung et al. [206] used a murine model of infection with Sendai virus and reasoned that CD49d/CysLTR1^+^ neutrophils stimulate the expression of Fc*ε*RI on conventional pulmonary DCs in a type I IFN-dependent manner. This results in a CD11b-driven interplay between neutrophils and DCs. Subsequently, Th1-biased anti-viral immunity transitions into Th2-driven atopic disease, with the development of a post-viral chronic asthma. Moreover, the life span of these pro-atopic neutrophils is extended through CysLTR1-mediated signaling [209]. This DC-neutrophil crosstalk was dependent on neutrophil populations expressing CD49d. Therefore, there is the potential to interrupt this pathogenesis by targeting CD49d^+^ neutrophils to prevent the development of post-infection atopic diseases. Interestingly, Cheung et al. [209] also identified neutrophils expressing CD49d/CycLTR1 in the nasal lavage of humans during acute respiratory viral infection, suggesting that the CysLTR1-dependent pathway identified in mice might apply to humans.

There is also a subset of neutrophils known as angiogenic neutrophils, which have a CD49d^+^, VEGFR1^high^, CXCR4^high^ phenotype and have been identified in humans and mice. These neutrophils are recruited from the circulation to hypoxic sites by vascular endothelial growth factor (VEGF)-A, where they promote angiogenesis via release of MMP-9. The use of neutralizing antibodies revealed that the recruitment of these neutrophils required VEGF-A, VEGF receptor-1, and VEGF receptor-2. The ability to block the recruitment of these neutrophils to areas of hypoxia and hinder revascularization using anti-CD49d therapy was also demonstrated. Furthermore, they showed that neutrophils express VEGF receptor-1, which can cause activation when stimulated with VEGF-A [210]. Therefore, developing therapies that target this subpopulation of neutrophils’ functional capacity for revascularization is beneficial in instances where control and regulation of angiogenesis would be valuable.

As part of the heterogeneity of neutrophils, there are fundamental differences in the functionality of NDNs and PMN-MDSCs. However, it is possible to induce a switch in subsets from NDNs to PMN-MDCSs through endoplasmic reticulum stress [122]. Therefore, the plasticity between NDNs and PMN-MDSCs can be manipulated though mechanisms that target endoplasmic reticulum stress. This could be an additional strategy to investigate in the future for the modulation of viral diseases in which these neutrophil subsets play a role.

Finally, it is worth mentioning that viral infections can modulate not only neutrophil heterogeneity but also neutrophil function to successfully establish infection. These can include hijacking neutrophils for replication (e.g., influenza virus), activating neutrophils to produce pro-inflammatory cytokines (e.g., SARS-CoV-2), or releasing neutrophil-derived granule products (e.g., HIV) [211,212,213,214]. Neutrophils represent a foundational immunological component responding to viral infections. Continuing to dissect their functions in conditions of health and disease promises to reveal even more opportunities for the development of novel, highly targeted therapies to prevent or reduce the clinical burden of virus-mediated diseases.

## 6. Conclusions

Neutrophils are important components of the immune system with great plasticity that are associated with undesirable off-target tissue damage as well as beneficial anti-viral responses (Figure 3). The difference between neutrophils providing a protective effect from a destructive effect, in most cases, appears to be linked to the duration of the responses as well as a loss of regulation of their responses. However, this may be due to specific subsets of neutrophils that tend to be resistant to immunoregulation. Moreover, the interplay of different micro-environmental signals can cause neutrophil heterogeneity. Indeed, the diversity of neutrophils both in homeostatic and disease conditions could be due to neutrophil responses to micro-environmental cues as well as a developmental programmed in neutrophil maturity. For instance, human neutrophils can function as anti gen-presenting cells through the upregulation of HLA-DR expression. However, not all neutrophils express HLA-DR, and not all stimuli induce the upregulation of HLA-DR. Therefore, using the term of neutrophil subsets in reference to differential phenotypes and functions must be avoided. Single cell analysis dissecting genetic alterations and molecular adaptability of the neutrophil populations in response to pathophysiological/immunological conditions could further define the phenotypic heterogeneity and functional versatility of neutrophils. The identification of phenotypes and markers that more reliably define neutrophil subpopulations during viral infections would allow for the development of new strategies to target these populations and modulate their anti-viral versus immune-mediated disease-promoting functions.

## Figures and Tables

**Figure 1 cells-11-01322-f001:**
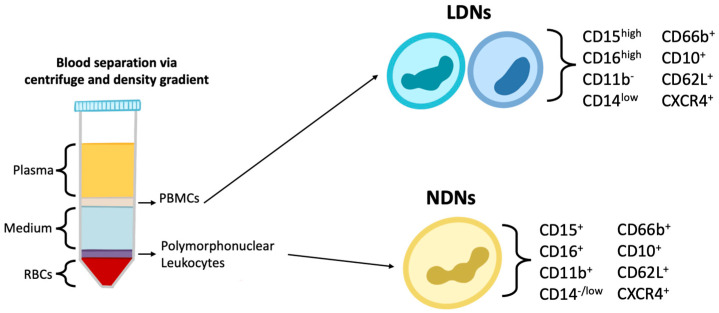
Morphologically distinct subsets of neutrophils are observed in circulation. Isolation of neutrophils from the blood by density gradient centrifugation will give two band layers of leukocytes. The lower band would have the polymorphonuclear cells and is where neutrophils are typically purified from. However, there are neutrophils found in the higher band that have lower density. Blood from healthy individuals was examined for these populations of neutrophils, commonly denoted as low density neutrophils (LDNs) and normal density neutrophils (NDNs). Differences in phenotypic surface markers were observed, as well as differences in their ability to produce reactive oxygen species and phagocytic capacities. Therefore, from blood-derived mononuclear cells, two different subsets can be detected and isolated based on density. Furthermore, these two subsets are functionally different and can be distinguished by flow cytometric assessment of surface markers [92].

**Figure 2 cells-11-01322-f002:**
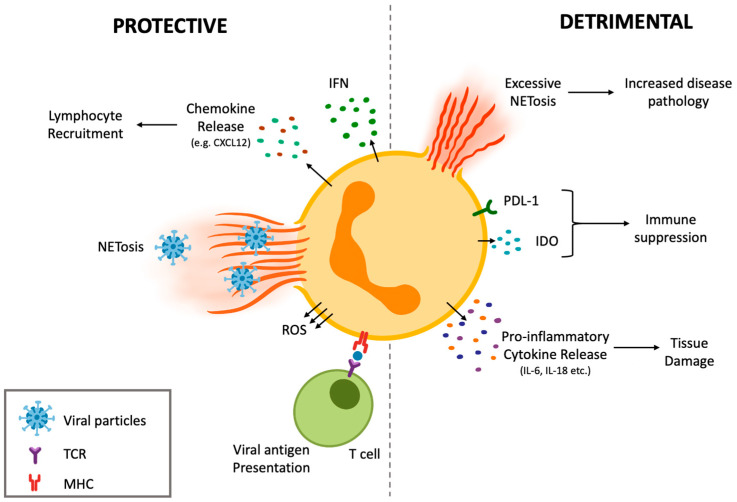
Neutrophils can be fine-tuned and influenced by their micro-environment to exert detrimental or protective immunological functions during anti-viral responses. Examples include recruitment of leukocytes through chemokine release and presentation of viral antigens to T cells. Additionally, neutrophils support type I interferon responses that mediate anti-viral immunity. However, neutrophils can also function in a manner that is harmful to the host. For example, neutrophils produce an array of pro-inflammatory cytokines, which when uncontrolled can lead to tissue damage. Similarly, when NETosis becomes dysregulated, there can be substantial bystander damage to tissues. Additionally, neutrophils have immunosuppressive capabilities that can hinder anti-viral responses, thereby reducing viral clearance.

**Figure 3 cells-11-01322-f003:**
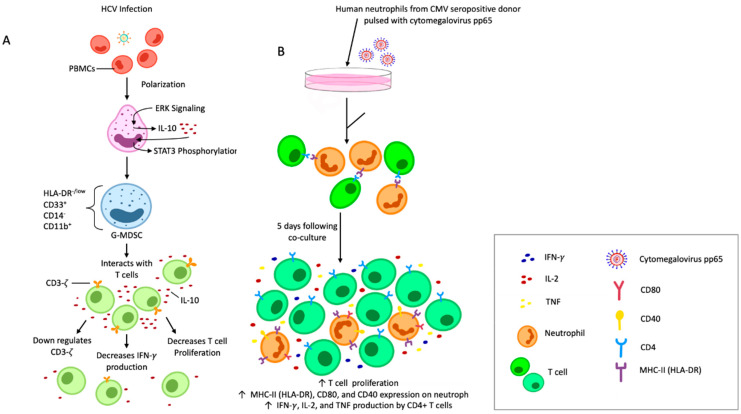
Neutrophils are heterogeneous, with each subset exerting specific immunomodulatory functions during viral infections such as those caused by hepatitis C virus (HCV). (**A**) Granulocyte myeloid-derived suppressor cell polarization occurs through activation of the extracellular signal-regulated protein kinase (ERK)-1/2 mitogen-activated protein kinase (MAPK) pathway, with further augmentation by IL-10-dependent signal transducer and activator of transcription (STAT)-3 phosphorylation. Eventually, the proliferation of autologous T cells and production of IFN-*γ* are repressed, leading to persistence of HCV [111]. (**B**) Human neutrophils can also function as antigen-presenting cells. Neutrophils pulsed with the cognate antigens pp65 from cytomegalovirus (CMV) or influenza hemagglutinin were able to present the antigens to autologous antigen-specific CD4^+^ T cells via major histocompatibility complex (MHC) class II and induce proliferation of helper T cells at a low but consistent rate both in vitro and ex vivo [139].

## Data Availability

Not applicable.

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
