# Peer review of "Neutrophil Functional Heterogeneity and Implications for Viral Infections and Treatments"

_cells, 2022, doi:10.3390/cells11081322_

Round 1
Reviewer 1 Report
The neutrophil is an important component of the immune system. Lily Chan et al reviewed the different phenotypes of neutrophils and their roles in viral infections, which is important for understanding Viral Infections and guiding precision treatment.
Points
1.There are not adequate references to related and previous work and some new references should be added in the manuscript.( for example: High-dimensional profiling reveals phenotypic heterogeneity and disease-specific alterations of granulocytes in COVID-19Proc Natl Acad Sci U S A. 2021 Oct 5;118(40):e2109123118. doi: 10.1073/pnas.2109123118.Increased Neutrophil Aging Contributes to T Cell Immune Suppression by PD-L1 and Arginase-1 in HIV-1 Treatment Naïve Patients.Front Immunol . 2021 Aug 18;12:670616. doi: 10.3389/fimmu.2021.670616. eCollection 2021.Preinfection laboratory parameters may predict COVID-19 severity in tumor patients Multicenter Study Cancer Med. 2021 Jul;10(13):4424-4436. doi: 10.1002/cam4.4023. Epub 2021 Jun 13.)
- There are some grammatical errors in the manuscript( such as “examined”(line 22) )and the sense should be past tense in the article.
Author Response
"Please see the attachment."

Reviewer 2 Report
These authors have written a review on neutrophil subsets. A discussion about linearity in neutrophil development or plasticity caused by local environmental cues is missing. The authors refer to a lot of papers and have gathered an enormous amount of data covering many aspects of ‘the neutrophil’ but it has remained unclear whether this is true for mice or humans or both.
Since murine and human neutrophil really differ in morphology, granule formation and content, as well as number wise, the review is difficult to apprehend and is actually incomplete. When mouse studies are referred to (which is often unclear throughout the text), we also need human studies to prove the concept, since this concept might not work in humans. This is - for instance - illustrated by the migration of neutrophils back into the bone marrow to get eliminated. Moreover, apoptotic neutrophils in humans do NOT respond to SDF1 or any other chemokine - making it hard to believe these cells go specifically to the bone marrow for their elimination. This concept has never been shown for humans and is still heavily debated if true at all. (And – by the way, neutrophils also do NOT make SDF1 by themselves.)
As another examples, the authors must rethink the theme of gMDSCs. This has become popular in the light of current COVID19 publications, and the role of neutrophils in viral disease in general has been given quite some thoughts. It has remianed unclear as to whether neutrophils only phagocytize intact viral particles, remnants, or are simply infected by the virus itself, or get activated to become gMDSCs, or otherwise. The discussion on g-MDSCs and LDNs is also rather superficial and uncritical. For instance, CD63 and CD66b are markers of neutrophil activation and may not serve to as ‘markers’ of low density unless the authors explain that density by granule release is the underlying response that LDNs arise in the circulation. And suddenly... we have three distinct LDN subsets? And if there are no surface markers to distinguish gMDSCs, how do we appreciate the activation status to obtain MDSC activity? Are these subsets, are these LDNs. How many neutrophil subsets do the authors finally think we have in our body and under which condition?
A critical appraisal is required (and not a mere anthology and biased selection of papers) to appreciate the complexity of neutrophil studies and the mere lack of solid data since comparisons in-between labs is often poor, as well as between species. Many references are reviews elsewhere, which is not the way a review should be written in the first place.
In sum, we miss a clear and analytical view which helps to develop a deeper understanding of the neutrophil, in its development, functions and plasticity.
Author Response
"Please see the attachment."

Reviewer 3 Report
Emerging evidences have shown that neutrophils play distinct roles during virus infection, impacting disease severity. Furthermore, single cell analyses have broken the concept that neutrophils are not plastic, which opens the possibility to elucidate new inflammatory mechanisms mediated by these cells.
Chan et al provides a review which summarizes several aspects of neutrophil immunobiology, which represents a precious overview for non-specialists. The description of neutrophils subtypes and their relevance in the context of virus infection is extremely relevant considering the covid-19 as well as future pandemics. Overall, I found the review well-written and the logical flow is well-organized. However, some addition could contribute to expand the role of neutrophil in viral infection and possible treatments.
1) It has been demonstrated that Gasdermin D is crucial for the process of NETosis (Sollberger et al Science Immunology 2018; Silva et al Blood 2021). This enzyme can be blocked by Disulfiram, a FDA-approved drug. Discussing the role of this enzyme during NETosis could add another option of treatment for covid-19, especially with a recent publication which demonstrate the efficacy of Disulfiram in protecting against covid-19 pathology (Adrover et al JCI Insight 2022).
2) NETs have been shown to induce type I IFN in the context of Systemic Lupus Erythematosus (Garcia-Romo et al Science Translational Medicine 2011). It is possible that NETs might have an indirectly anti-viral role by inducing type I IFN.
3) Treatment with dexamethasone in covid-19 patients has been shown to modulate neutrophils into immature neutrophils with immunosuppressive properties (Sinha et al Nature Medicine 2021). It fits with the authors discussion of anti-inflammatory neutrophils.
4) the review covers a considerable amount of evidences describing neutrophils heterogeneity. Nevertheless, making tables or figures to summarize the information helps with the manuscript understanding.
Author Response
"Please see the attachment."

Round 2
Reviewer 1 Report
Lily Chan et al discussed the different phenotypes of neutrophils, their roles in viral infections and the possible ways to target neutrophil subsets during viral infections as potential anti-viral treatments, which is to facilitate a better understanding of neutrophil functionality and polarization during inflammation and on our ability to treat different inflammatory reactions.
Author Response
We appreciate the reviewer's comments.
Reviewer 2 Report
Major concern:
The authors have adapted their review on neutrophil subsets in viral disease which has clearly improved but at some points still rather imprecise and incomplete. It is not the huge number of references that count but the content that matters, as indicated below.
First and again, a separate discussion about linearity in neutrophil development or plasticity caused by local environmental cues is still missing at the beginning of the text. The reader would be helped to know what the stages of development and functional maturation (PMID: 23650620, PMID: 27558325; PMID: 30630937; PMID: 28263321; PMID: 31747616) exactly represent. Are we talking about separate developmental subsets or is it local influences that in part influences the behavior of these cells? A brief outline at the start will help to address the plasticity with which the authors finally end their review (in particular when discussing senescence and the debate about MDSC subsets).
Second and foremost, we need a separate subheading on the differences between the two main species studied. The authors refer to ‘the neutrophil’ as if mouse and man are identical. I would like to have it clearly indicated throughout the text when the data are truly human or where purely mouse data. I suggest a clear and separate paragraph on the differences between human and mouse neutrophils in their phenotype, morphology, phenotype, granule composition, functional differences etc etc. These neutrophils are NOT identical in many aspects.
Although improved, the lack of clarity makes this review still rather shallow with a lot of details but no real depth and discussion. Little news after all and no firm statements on what is lacking (gaps of knowledge). For instance, the subjects covered by the authors like reverse migration by Nourshargh’s group and others has never been shown in humans. This might still be a murine phenomenon only and an experimental ‘artifact’ of little relevance in human conditions. Same is true for the bone marrow as a ‘graveyard’ or senescent neutrophils and the role of SDF1 in homing to this place to ‘vanish’ after a short lifespan in the circulation. Maybe the authors want to have the reader believe that the neutrophils patrol the body like DCs and re-enter the blood and end up by cues to die in the bone marrow? Then we need evidence in humans as well. As a final example, I refer to the data of Hampton et al on neutrophil migration to lymph nodes. Interesting that neutrophils are efficiently recruited to sites of both microbial and sterile lesions, subsequent re-localization to draining lymph nodes happens only when bacteria are present in the primary lesion. Skin egress of neutrophils occurs via lymphatic vessels and is dependent on CD11b and CXCR4. But human neutrophils do not migrate towards SDF1 like mouse cells, how young or old the human neutrophils might be (in contrast to ref 56, see e.g. PMID: 17379064).
When evidence in humans is not available, the authors should indicate this very clearly and not confuse the readers where there is already a lot of ‘mist’ around the theme of neutrophil subsets. The lack of human data to support the claims in mouse studies leaves us with much of a scientific debate about the actual meaning of many findings in mice for humans. A critical challenge of these murine data and uncritical acceptance of these data for human neutrophils is missing.
Additional concerns:
The statement on generating CD15+ LDNs (line 742) is highly suggestive and in practice highly impractical for many reasons. How would the authors suppose to manage the production of large quantities of such cells? Without a clear idea on the practicalities of generating primary neutrophils, the statement is a rather meaningless remark that the authors better could omit.
The distinction between G-MDSCs and PMN-MDSCs is vague and remains undefined (if a distinction between the two is correct at all).
The discussion on LDNs have still not become clear to me. In healthy individuals these cells may not exist and hence the paragraph starting at line 307 emphasizes a single paper which may be an artifact from a lab working with preactivated human blood samples (taken from healthy individuals). As every neutrophil aficionado knows, these cells are delicate and easily activated or primed during transport, purification and further handling during washing steps, etc.
In addition, I wonder what evidence the authors refer to for humans when discussing young myeloid G-MDSCs. Are these cells a separate immature subset and if so what data on cellular activity truly indicate that these cells exert MDSC activity? If the authors believe this G-MDSC subset of neutrophil exists on the basis of solid data, this should be linked to the bone marrow developmental stages of human neutrophils and monogenic diseases that would lead to a defect in this subset of G-MDSC cells (see above).
The suggestion made at line 410 suggests that PMN-MDSCs and neutrophils are different entities. On which papers do the authors base the statement? I would suggest to read PMID: 30730851 and PMID: 31738831. Are these entities really different or is the literature highly biased towards mouse studies and clearly not supported by the current neutrophil studies in humans?
The reports on HLA-DR expression by neutrophils are important to support the plasticity of (human) neutrophils and role as APC in human disease. The authors might focus a little more on this subject because also here there is considerable controversy but relevant in the context of viral disease (Figure 3). Also read PMID: 7545027, PMID: 9166855, PMID: 28143882, PMID: 31182561 as further supportive evidence for the expression of HLA-DR on a fraction of neutrophils. This is only possible under certain conditions and is definitely context-dependent (i.e. plasticity by environmental conditioning). Adding these additional data on HLA-DR expression by a fraction of human neutrophils and only under certain specific conditions with potential APC activity of human neutrophils convincingly broadens the context of antigen-presenting neutrophil activity in viral disease (Figure 3). Importantly, not all stimuli and not all neutrophils will become positive for HLA-DR expression – at least as far as the conditions tested in vitro and in vivo seem to indicate to date. We must remember that - even when a fraction of the cells is found to become positive – such findings do not prove it to be a specific cellular subset. Taking the remainder of HLA-DR negative neutrophils and activate these under similar conditions again in a subsequent culture could potentially result again in a similar fraction of positive HLA-DR-expressing cells. Such experiments are lacking (and difficult to perform due to the short lifespan of neutrophils) but clearly underscore the need to remain critical on the use of the word ‘subset’ when neutrophil plasticity is meant instead of the suggestion of a separate, phenotypically and/or functionally well defined developmental cell type.
finally, there are minor spelling errors, e.g. C-CSF for G-CSF, CECAM for CEACAM, and also some conceptual flaws. For instance, the sialylated Lewis-x moieties instead of CD15 constitute the ligands for selectins.
Reviewer 3 Report
N/A
Author Response
We thank the reviewer for the recommendations.